

# Agreement and differences between the equations for estimating muscle and bone mass using the anthropometric method in recreational strength trainees

Nicolás Baglietto[1,*], Mario Albaladejo-Saura[1,2], Francisco Esparza-Ros[1] and Raquel Vaquero-Cristóbal[3,*]

[1] Sport Injury Prevention Group, International Chair of Kineanthropometry, UCAM Universidad Católica de Murcia, Murcia, Spain
[2] Sport Injury Prevention Group, Facultad de Deporte, UCAM Universidad Católica de Murcia, Murcia, Murcia, Spain
[3] Department of Physical Activity and Sport Sciences, Faculty of Sport Sciences, University of Murcia, San Javier, Spain
* These authors contributed equally to this work.

Corresponding author
Mario Albaladejo-Saura,
mdalbaladejosaura@ucam.edu

## ABSTRACT

**Introduction:** The interest in estimating muscle mass (MM) and bone mass (BM) has grown in the sporting arena, and more specifically in recreational strength trainees, leading to the creation of different strategies to assess them. The aims were: 1) to investigate the agreement between different MM and BM formulas, and the muscle-bone index (MBI), and to establish the differences between them, in a healthy young adult population; and 2) to analyze if there are differences between males and females in the comparison of MM, BM and MBI formulas.

**Methods:** This study followed a descriptive cross-sectional design. A total of 130 adult active recreational strength trainees were evaluated according to the procedures described by the International Society for the Advancement in Kinanthropometry (ISAK). Estimations were made in kilograms of MM and BM by following the equations by different authors.

**Results:** The results showed significant differences between the values obtained by all the MM and BM formulas in the general sample ($p < 0.001$), and by the majority of formulas for male and female samples. In the general sample, Lin's coefficient indicated a strong agreement between Kerr, Lee, and Poortmans' MM estimation equations (concordance correlation coefficient (CCC) = 0.96–0.97). However, when stratifying by sex, this agreement persisted only in males (CCC = 0.90–0.94), in contrast with a lack of agreement observed in females (CCC < 0.90). Discrepancies in bone mass agreement were noted both in the general sample (CCC < 0.15) and when stratified by sex (CCC < 0.12).

**Conclusions:** In general, differences were found between the values reported by the MM and BM formulas in recreational strength trainees, without an agreement between them. Sex was shown to significantly influence the differences found. The practical implications are that when comparing an individual with reference tables, other studies, or if analyzing an individual's evolution, the same estimation equation should be used, as they are not interchangeable.

# INTRODUCTION

Higher muscle mass (MM) has been commonly associated with greater athletic performance in most sports (*Lockie et al., 2021*). In turn, bone mass (BM) provides structure and support to the MM. Along this line, an adequate development of BM will provide a favorable environment for a greater development of MM (*Esparza-Ros & Vaquero-Cristóbal, 2023*). In addition, the estimation of MM and BM has also been widely used in the field of health, because of its relationship with diseases such as sarcopenia, osteopenia, and osteoporosis, or quality of life, especially in elders (*Mosti et al., 2013*; *Clynes et al., 2021*; *Mastavičiūtė et al., 2021*). Due to this, in recent years the research of methods for the estimation of muscle and bone masses has been in the spotlight of the scientific community (*Kasper et al., 2021*).

Anthropometry has been widely utilized through the use of different methods to estimate body composition, especially in the clinical setting, due to their simplicity, low cost and high validity and reliability (*Costa-Moreira et al., 2015*). This approach has certain advantages, such as being little affected by factors that affect body water or food intake, facilitating its replicability (*Costa-Moreira et al., 2015*; *Kerr, Slater & Byrne, 2017*; *Zambone, Liberman & Garcia, 2020*). In fact, this approach has been proposed as the ideal one to be used in cases when it is not possible to ensure the maintenance of standard measurement conditions (*Kasper et al., 2021*). Therefore, anthropometry is usually the most common indirect method for the estimation of MM and BM in clinical and sports settings, due to its advantages in field studies (*Costa-Moreira et al., 2015*).

However, anthropometry is not free of problems, with the main one being the large number of equations that exist for the estimation of MM and BM, and for relating these two parameters (*Esparza-Ros & Vaquero-Cristóbal, 2023*).

Previous studies have analysed the differences in fat mass formulas with anthropometry, finding that they are not interchangeable, which could be due to divergences in the variables included in each of these formulas, the populations in which they were validated, the method used as the gold standard for their validation, or the measurement protocols followed (*Vaquero-Cristóbal et al., 2020*; *Mecherques-Carini et al., 2022*). These studies that have analysed the interchangeability of fat mass formulas with anthropometry have also found that sex could affect this interchangeability (*Mecherques-Carini et al., 2022*), as a consequence of the sexual dimorphism that occurs after puberty (*Wells, 2007*). Nevertheless, to date, only one study has explored the interchangeability of muscle mass (MM) formulas with anthropometry (*Fernández-Vieitez & Aguilera, 2001*). This study suffers from several limitations, such as the use of a small sample of only 20 subjects, failure to specify the participants' sex, and the exclusivity of the sample to Cuban weightlifters, raising questions about the generalizability of the results to populations with different characteristics. Additionally, given the recognized sexual dimorphism in MM and MO, the lack of consideration of sex may distort the comparison between the anthropometric equations used to estimate these masses (*Bredella, 2017*). It is worth noting that the

influence of sex has not been addressed in previous research. Furthermore, no previous study has examined the agreement between different equations for estimating BM using anthropometry or between different muscle/bone mass ratios (MBI).

Therefore, the aims of the present study were: 1) to investigate the agreement between different MM and BM formulas, and the MBI, and to establish the differences between them, in a healthy young adult population; and 2) to analyze if there are differences between males and females in the comparison of MM, BM and MBI formulas. Building upon the insights gleaned from prior research, our research hypotheses were as follows: a) the different formulas for estimate MM, BM and MBI with anthropometry are not interchangeable; and b) sex exerts influence on the comparative analysis of these formulas. This study is part of the doctoral thesis of Nicolás Baglietto.

## MATERIALS AND METHODS

### Participants

The calculations necessary to establish the sample size were performed with Rstudio 3.15.0 software (Rstudio Inc., Boston, MA, USA). The significance level was set at $\alpha = 0.05$. The standard deviation (SD) was established based on the percentage of MM from previous studies (SD = 1.58) (*Sellés-Pérez et al., 2019*). With an error (d) in the corrected arm circumference of 0.39 cm, the required sample was 59 subjects per group. Therefore, a total of 130 Caucasian subjects participated in the present study, of whom 71 were male (mean age = $26.52 \pm 6.01$ years, mean body mass = $75.66 \pm 7.81$, mean stretch stature = $174.73 \pm 6.15$ cm), and 59 female (mean age = $27.03 \pm 6.19$ years; mean body mass: $59.02 \pm 7.38$ kg, mean stretch stature = $161.87 \pm 5.6$ cm).

### Study design

A prospective descriptive cross-sectional design was followed, in accordance with the Strengthening the Reporting of Observational studies in Epidemiology (STROBE) guidelines for observational studies (*Cuschieri, 2019*). The ethics committee of the Catholic University of Murcia reviewed and authorized the protocol designed for data collection according to the World Medical Association code (number CE072103). During the process, the statements of the Helsinki declaration were followed. All the participants were informed about the procedures and signed voluntarily by written an informed consent form before the start of the study where they gave their consent to participate in the present investigation.

### Selection criteria

The inclusion criteria were: A) aged between 18 and 45 years old; B) Caucasian (*Esparza-Ros et al., 2022*); C) physically active individuals according to the World Health Organization (WHO) (*Bull et al., 2020*); D) having a minimum of 2 years of uninterrupted sports experience performing planned strength training, in at least two sessions per week lasting 1 h per session (*American College of Sports Medicine position stand, 2009*); and E) in the case of women, being between days 8 and 21 of the menstrual cycle at the time of the assessment (*Esparza-Ros et al., 2022*). The exclusion criteria were: A) having any injury or

pathology that would condition the measurements; B) having performed vigorous exercise within the 24 h prior to the measurement session or moderate exercise in the 12 h prior to the measurement session; C) having performed physical exercise on the same day as the measurements were taken; D) having consumed products with diuretic properties or having eaten a heavy meal in the 24 h prior to the study (*Kerr, Slater & Byrne, 2017*); E) consuming an ergogenic aid or sports supplement at the time of the evaluations (*Pakulak et al., 2021*); F) hormonal or corticosteroid treatment, excluding menstrual cycle regulation treatment, in the 3 months prior to the evaluation (*Esparza-Ros et al., 2022*); G) following a specific dietary regimen (*da Godois et al., 2020*); H) presenting dehydration or hyperhydration at the time of the test; I) not being in a state of fasting since the night before the evaluation (*Kerr, Slater & Byrne, 2017*); J) presenting any disease that could affect the estimation of body composition with anthropometry (*Esparza-Ros et al., 2022*); and K) not keeping the appointment for the anthropometric assessment.

## Procedure

A full-profile anthropometric assessment was performed, following the guidelines from the International Society for the Advancement of Anthropometry (ISAK) (*Esparza-Ros, Vaquero-Cristóbal & Marfell-Jones, 2019*), to subsequently estimate MM with four different formulas, BM with two different formulas, and MBI with four different formulas.

Volunteers were recruited through targeted advertisements on social media, press releases, and local sports facilities in the La Plata region, Argentina. Interested individuals were required to complete an initial form providing basic information. Those meeting inclusion/exclusion criteria were then scheduled for anthropometric assessments. The flow diagram for the sampling is shows in Fig. 1.

Prior to the measurements, the participants were asked to avoid heavy meals, diuretic products, and physical activity starting from the day before the measurement session, to avoid physical activity on the day of the measurements, to follow normal hydration guidelines, and to fast starting from the night before the measurements were taken. For each subject, the full set of anthropometric measurements were performed in a single day, from 7 am to 11 am, in a private room with a comfortable, standardized temperature (*Esparza-Ros, Vaquero-Cristóbal & Marfell-Jones, 2019*).

First, the participants self-completed a sociodemographic questionnaire in which information was collected about their sex, age, diseases, corticosteroid hormone treatment, specific dietary regimes, physical exercise practice in the last 24 h, consumption of ergogenic aids or dietary supplements, and menstrual cycle periodization in the case of women, and their hydration status was checked through urine osmolarity, using protocols described in previous research (*Esparza-Ros et al., 2022*).

After this, the anthropometric variables were evaluated. Two basic measurements (height and body mass), four skinfolds (triceps, subscapular, thigh and calf), eight girths (head, arm relaxed, arm flexed and tensed, forearm, chest, thigh 1 cm gluteal, thigh middle, and calf), and seven breadths (biacromial, biilliocristal, transverse chest, humerus, bistyloid, femur, and bimalleolar), were carried out according to the guidelines from the International Society for the Advancement in Kinanthropometry (ISAK) (*Esparza-Ros,*

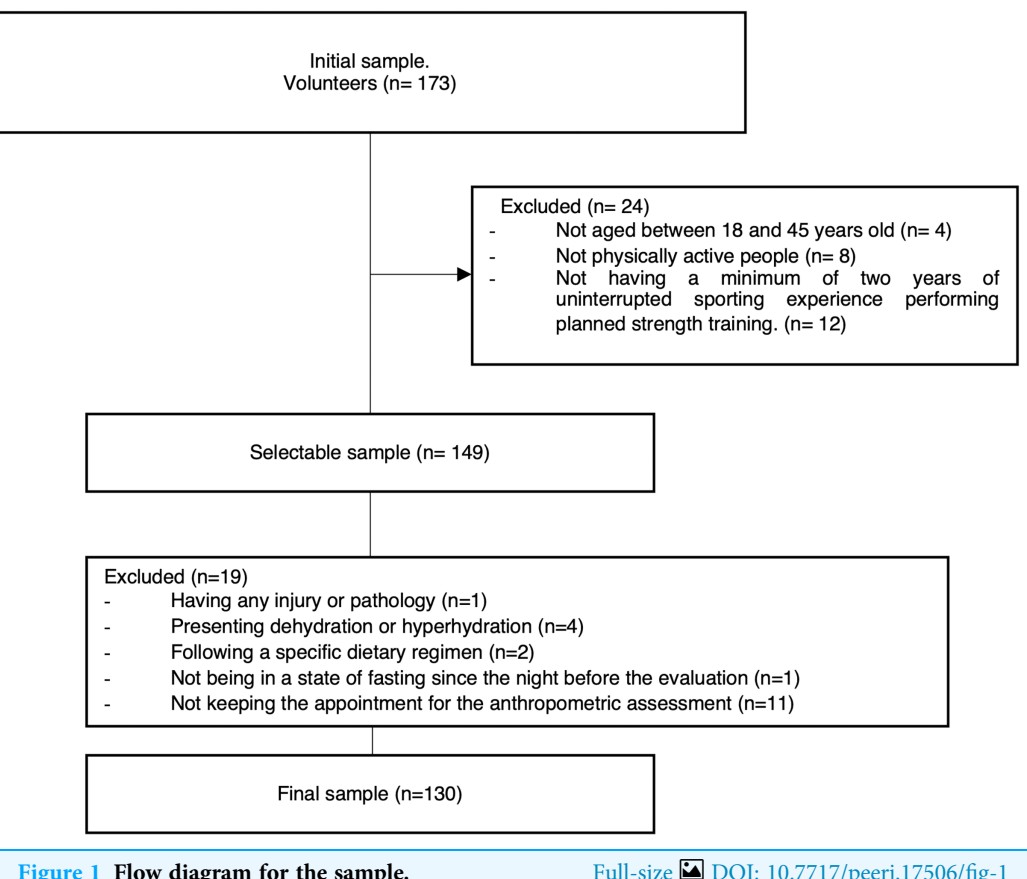

**Figure 1  Flow diagram for the sample.**   

*Vaquero-Cristóbal & Marfell-Jones, 2019*) by an ISAK-accredited level 3 anthropometrist. Each measurement was performed twice. If the difference between the measurements was greater than 5% in skinfolds and 1% in the rest of measurements between the two measurements, a third evaluation was performed. The final value for data analysis was the mean when two measurements were needed, or the median in cases where three measurements were taken. The intra-evaluator technical error of measurement was 0.01% for basic measurements, 0.9% for skinfolds, 0.6% for girths, and 0.4% for breadths.

A SECA 862 weighing scale (SECA, Hamburg, Germany) with an accuracy of 0.1 kg was used to measure body mass, and a SECA 213 portable stadiometer (SECA, Hamburg, Germany) was used to measure stretch stature. The skinfolds thicknesses were measured with a Harpenden skinfold caliper (British Indicators, Surrey, UK). Girths were measured with an inextensible anthropometric Lufkin W606PM tape measure (Lufkin, New York, NY, USA), and breadths with a large sliding caliper and a small sliding caliper (Rosscraft, Argentina). All the equipment was calibrated before taking measurements to avoid possible causes of error in the measurements.

Subsequently, the kilograms and percentages of MM were estimated according to the formulas from *Lee et al. (2000)*, *Poortmans et al. (2005)*, *Matiegka (1921)* and *Kerr & Ross (1991)*. The kilograms and percentages of BM were also estimated according to the formulas from *Rocha (1975)* and *Kerr & Ross (1991)*. All of these formulas had been

**Table 1 Equations for the estimation of muscle mass, bone mass and muscle/bone index included in the study for general sample and divided by sex.**

| Formulas | Author | Specific formula |
|---|---|---|
| Muscle mass (kg) | Kerr | Muscle mass (kg) = [(Z-score muscle * 4.4) + 24.5]/(170.18/Height in cm)$^3$; where: Z-score muscle = [(Sum of corrected girths in cm * (170.18/Height in cm) − 207.21)/13.74]; and Sum of corrected girths in cm = (Relaxed arm girth in cm-π * Triceps skinfold in cm) + Forearm girth in cm + (Chest girth in cm-π * Subscapular skinfold in cm) + (Thigh middle girth in cm-π * Thigh skinfold in cm) + (Calf girth in cm-π * Calf skinfold in cm) |
| | Lee | Muscle mass (kg) = Height in m * [0.00744 * (Relaxed arm girth in cm-π * Triceps skinfold in cm)$^2$ + 0.00088 * (Thigh middle girth in cm-π * Thigh skinfold in cm)$^2$ + 0.00441 * (Calf girth in cm-π * Calf skinfold in cm)$^2$] + 2.4 * (1 if male; 0 if female) − 0.048 * Age in years + (1 if black race; 0 if Caucasian race; and −1 if Asian race) + 7.8 |
| | Poortmans | Muscle mass (kg) = Height in m * [0.0064 * (Relaxed arm girth in cm-π * Triceps skinfold in cm)$^2$ + 0.0032 * (Thigh middle girth in cm-π * Thigh skinfold in cm)$^2$ + 0.0015 * (Calf girth in cm-π * Calf skinfold in cm)$^2$] + 2.56 * (1 if male; 0 if female) + 0.136 * Age in years |
| | Matiegka | Muscle mass (kg) = Height in m * [[(Arm relaxed girth in cm/π-Triceps skinfold in cm) + (Thigh middle girth in cm-π * Thigh skinfold in cm) + (Calf girth in cm-π * Calf skinfold in cm) + (Chest girth in cm-π * Subscapular skinfold in cm)]$^{0.125}$]$^2$ * 6.41 |
| Bone mass (kg) | Kerr | Bone mass in kg = Bone mass of the body in kg + Bone mass of the head in kg; where: Bone mass of the body in kg = (Z-score body bone * 1.34 + 6.7)/(170.18 / Height in cm$^3$); Z-score body bone = [Sum of Breadths in cm * (170.18/Height in cm) − 98.88] / 5.33; Sum of Breadths in cm = Biacromial breadth in cm + Biiliocristal breadth in cm + (Humerus breath in cm * 2) + (Femur breadth in cm * 2); Bone mass of head in kg = (Z-score head bone * 0.18) + 1.2; and Z-score head bone = [Head girth in cm * (170.18/Height in cm) − 56]/1.44 |
| | Rocha | Bone mass in kg = 3.02 * (Height in m$^2$ * Bi-styloid breadth in m * Femur breadth in m * 400)$^{0.712}$ |
| Muscle/Bone index | Kerr/Kerr | Muscle mass with Kerr formula/Bone mass with Kerr formula |
| | Lee/Rocha | Muscle mass with Lee formula/Bone mass with Rocha formula |
| | Poortmans/Rocha | Muscle mass with Poortmans formula/Bone mass with Rocha formula |
| | Matiegka/Rocha | Muscle mass with Matiegka formula/Bone mass with Rocha formula |

validated in a population with characteristics similar to those in the present study. Once the values of both MM and BM were obtained, the MBI was calculated: MBI = MM (kg)/BM (kg) (*Esparza-Ros & Vaquero-Cristóbal, 2023*) with all possible equation relations. The formulas used to estimate MM, BM and MBI were the same for the general sample as for the male and female sample. In summary, Table 1 shows the list of formulas used for the estimation of MM, BM and MBI both for the general sample and segmented by sex.

## Patient and public involvement

The patients and the public were not involved in the development of the research questions, the design, or performance of the study. However, participants were involved in the recruitment of others through the snowball method. The study results were shared with the participants with an individual report and will be shared with other relevant stakeholders through various social media handles and conferences after the publication of the paper.

## Statistical analysis

The normality of the distribution was tested with the Kolmogorov-Smirnov test. The kurtosis analysis showed a platykurtic distribution for all variables. All the variables included in the analysis showed a normal distribution, so parametric statistical tests were performed. Descriptive statistics were performed for all variables. The agreement between equations was determined using Lin's concordance correlation coefficient (CCC), including precision (ρ) and accuracy (Cb) indexes, as well as with McBride's strength concordance (almost perfect >0.99; substantial >0.95 to 0.99; moderate = 0.90–0.95; and poor <0.90), following previous research (*Esparza-Ros et al., 2022*). Differences between MM, BM and MBI equations were obtained with a one-way analysis of variance (ANOVA) for repeated measurements. The effect size was calculated with partial eta squared ($\eta_p^2$). The Bonferroni *post hoc* adjustment was used to analyze differences between groups when these differences were significant. The significance level was set at $p = 0.05$. The software used to perform the statistical analysis was SPSS, in the case of the descriptive analysis and the ANOVA (v.23, IBM, Endicott, NY, USA), and MedCalc Statistical Software in the case of Lin´s concordance correlation coefficient (v.20.106, Mariakerke, Belgium).

## RESULTS

The descriptive analysis of MM, BM, and MBI calculated by the different authors are shown in Table 2 for the general sample and the male and female sample.

When analyzing the ANOVA results (Table 3), it was found that all the formulas for estimating MM and BM in kg and percentage, and MBI, showed significant differences between them ($F = 212.57$–$657.99$; $p < 0.001$; $\eta_p^2 = 0.657$–$0.837$). The covariable analysis revealed a significant effect of sex on the differences between formulas for all parameters analyzed ($F = 23.56$–$212.57$; $p < 0.001$; $\eta_p^2 = 0.155$–$0.624$), except for the percentage of bone mass ($F = 0.000$; $p = 0.996$; $\eta_p^2 = 0.000$) (Table 3).

When the pairwise comparison was performed for the general sample (Table 4), a significant difference was observed between all formula pairs of MM in kg ($p \le 0.001$) and as a percentage ($p < 0.05$); of BM in kg ($p < 0.001$) and percentage ($p < 0.001$); and of MBI ($p < 0.001$). In the pairwise comparison within the male group (Table 4), considering MM in kilograms, significant differences were found in all the pairs ($p < 0.05$), except for Lee and Poortmans ($p = 1.000$), and Poortmans and Matiegka ($p = 0.29$). Regarding MM as a percentage, significant differences were found in all the pairs ($p < 0.05$), except for Lee and Matiegka ($p = 0.145$). Concerning BM, significant differences were found in kg and percentage ($p < 0.001$). With regard to MBI, significant differences were found in all the pairs ($p < 0.001$), except for Lee/Rocha *vs.* Poortmans/Rocha ($p = 1.000$), Lee/Rocha *vs.* Matiegka/Rocha ($p = 0.087$), and Poortmans/Rocha *vs.* Matiegka/Rocha ($p = 0.611$). In the female group (Table 4), MM in kg and percentage showed significant differences between all the formula pairs ($p < 0.001$), except Kerr and Lee ($p = 1.000$). Regarding BM, significant differences were found in kg and percentage ($p < 0.001$). Lastly, for MBI, significant differences were also found between all the indices ($p < 0.01$).

**Table 2 Descriptive analysis of muscle and bone variables and muscle-bone index in the general sample and divided by sex.**

| Variable | Formula | General sample ($n = 130$) | | Male sample ($n = 71$) | | Female sample ($n = 59$) | |
|---|---|---|---|---|---|---|---|
| | | Mean ± SD | Max.–Min. | Mean ± SD | Max.–Min. | Mean ± SD | Max.–Min. |
| MM (kg) | Kerr | 28.50 ± 7.26 | 45.54–15.96 | 33.74 ± 5.24 | 45.55–20.39 | 22.21 ± 3.20 | 28.64–15.97 |
| | Lee | 27.93 ± 6.33 | 41.57–16.40 | 32.88 ± 3.84 | 41.57–23.08 | 21.98 ± 2.35 | 26.64–16.41. |
| | Poortmans | 27.37 ± 6.86 | 41.94–15.52 | 32.66 ± 4.11 | 41.94–22.00 | 21.02 ± 3.04 | 29.86–15.53 |
| | Matiegka | 30.33 ± 2.17 | 34.67–25.96 | 31.89 ± 1.39 | 34.68–28.85 | 28.45 ± 1.25 | 31.14–25.97 |
| MM (%) | Kerr | 41.38 ± 5.15 | 53.10–27.97 | 44.49 ± 4.41 | 53.10–28.23 | 37.65 ± 3.09 | 43.88–27.98 |
| | Lee | 40.76 ± 4.42 | 50.37–27.52 | 43.51 ± 3.30 | 50.37–31.92 | 37.46 ± 3.16 | 44.81–27.53 |
| | Poortmans | 39.79 ± 5.20 | 50.57 – 28.05 | 43.20 ± 3.74 | 50.58–29.66 | 35.69 ± 3.48 | 43.53–28.06 |
| | Matiegka | 45.31 ± 5.02 | 59.17 – 36.21 | 42.47 ± 3.41 | 52.25–36.21 | 48.73 ± 4.51 | 59.18–39.32 |
| BM (kg) | Kerr | 7.77 ± 1.45 | 11.17–5.04 | 8.75 ± 1.12 | 11.18–5.37 | 6.59 ± 0.80 | 8.76–5.04 |
| | Rocha | 10.64 ± 1.79 | 14.52–7.06 | 11.94 ± 1.20 | 14.52–8.97 | 9.08 ± 0.92 | 10.71–7.07 |
| BM (%) | Kerr | 11.42 ± 1.05 | 6.81–14.37 | 11.59 ± 1.13 | 14.37–6.81 | 11.22 ± 0.93 | 13.51–9.43 |
| | Rocha | 15.68 ± 1.39 | 19.44–12.06 | 15.85 ± 1.47 | 19.45–13.01 | 15.48 ± 1.28 | 18.33–12.07 |
| MBI | Kerr/Kerr | 3.63 ± 0.44 | 4.70–2.71 | 3.86 ± 0.39 | 4.70–2.92 | 3.37 ± 0.35 | 4.31–2.71 |
| | Lee/Rocha | 2.60 ± 0.29 | 3.43–1.87 | 2.76 ± 0.28 | 3.44–2.17 | 2.43 ± 0.20 | 2.98–1.88 |
| | Poortmans/Rocha | 2.55 ± 0.36 | 3.66–1.74 | 2.74 ± 0.32 | 3.66–2.13 | 2.32 ± 0.26 | 3.36–1.74 |
| | Matiegka/Rocha | 2.89 ± 0.30 | 3.67–2.29 | 3.15 ± 0.22 | 3.67–2.71 | 2.69 ± 0.19 | 3.23–2.30 |

**Note:**
SD, standard deviation; Max, maximum value; Min, minimum value; MM, Muscle Mass; BM, Bone Mass; MBI, Muscle/Bone index; kg, Kilograms; %, Percentages.

**Table 3 Differences in muscle and bone variables and muscle-bone index.**

| | ANOVA | | | ANCOVA | | |
|---|---|---|---|---|---|---|
| Variable | F | $p$ | $\eta_p^2$ | F | $p$ | $\eta_p^2$ |
| MM (kg) | 244.84 | 0.000 | 0.657 | 182.62 | 0.000 | 0.588 |
| MM (%) | 321.78 | 0.000 | 0.715 | 212.57 | 0.000 | 0.624 |
| BM (kg) | 212.57 | 0.000 | 0.808 | 23.56 | 0.000 | 0,155 |
| BM (%) | 657.99 | 0.000 | 0.837 | 0.000 | 0.996 | 0.000 |
| MBI | 375.19 | 0.000 | 0.745 | 147.22 | 0.000 | 0.534 |

**Note:**
F, ANOVA analysis value; $\eta_p^2$, eta-squared value; MM, Muscle Mass; BM, Bone Mass; MBI, Muscle/Bone index; kg, Kilograms; %, Percentages.

Table 5 shows the agreement between the formulas used to estimate MM, BM, and the MBI. Concerning the general sample, it is noteworthy that most equations displayed a poor concordance coefficient (CCC < 0.90; r = −0.37 to 0.91; Cb = 0.49–0.98), except in MM in kilograms for Kerr, Lee, and Poortmans (CCC = 0.96–0.97; r = 0.97; Cb = 0.98–0.99). For the male group, a similar trend was observed, where most equations showed a concordance coefficient below 0.90 (CCC < 0.90; r = −0.04 to 0.87; Cb = 0.44–0.99), with the exception of MM in kilograms for Kerr, Lee, and Poortmans (CCC = 0.95–0.97; r = 0.95 to 0.97; Cb = 0.94–0.99), and MBI for Lee/Rocha and Poortmans/Rocha (CCC = 0.92; r = 0.94; Cb = 0.99). For the female group, a poor concordance coefficient was evident among all estimation formulas (CCC < 0.90; r = −0.16 to 0.91; Cb = 0.11–0.99).

**Table 4 Differences between the formulas for muscle mass, bone mass and muscle/bone index for the general sample and divided by sex.**

| Variable | Formula | | General sample (n = 130) | | | Male sample (n = 71) | | | Female sample (n = 59) | | |
|---|---|---|---|---|---|---|---|---|---|---|---|
| | | | Mean differences ± SD | p value | 95%CI (min;max) | Mean differences ± SD | p value | 95%CI (min;max) | Mean differences ± SD | p value | 95%CI (min;max) |
| MM (kg) | Kerr | Lee | 0.57 ± 0.14 | 0.001 | [0.17–0.97] | 0.86 ± 2.18 | 0.001 | [0.27–1.45] | 0.22 ± 1.91 | 1.000 | [−0.29 to 0.74] |
| | Kerr | Poortmans | 1.12 ± 0.14 | 0.000 | [0.75–1.50] | 1.08 ± 0.20 | 0.000 | [0.53–1.64] | 1.18 ± 1.90 | 0.000 | [0.67–1.71] |
| | Kerr | Matiegka | −1.82 ± 0.47 | 0.001 | [−3.08 to −0.56] | 1.84 ± 5.10 | 0.003 | [0.46–3.23] | −6.24 ± 2.99 | 0.000 | [−7.06 to −5.43] |
| | Lee | Poortmans | 0.55 ± 0.12 | 0.000 | [0.21; 0.89] | 0.22 ± 1.59 | 1.000 | [−0.21 to 0.65] | 0.96 ± 1.91 | 0.000 | [0.44–1.48] |
| | Lee | Matiegka | −2.39 ± 0.38 | 0.000 | [−3.43 to 1.36] | 0.98 ± 3.39 | 0.030 | [0.06–1.90] | −6.47 ± 1.97 | 0.000 | [−7.01 to −5.93] |
| | Poortmans | Matiegka | −2.95 ± 0.43 | 0.000 | [−4.12 to −1.78] | 0.76 ± 3.82 | 0.298 | [−0.27 to 1.80] | −7.43 ± 2.96 | 0.000 | [−8.24 to −6.62] |
| MM (%) | Kerr | Lee | −0.62 ± 0.21 | 0.027 | [0.47–1.20] | 0.99 ± 2.83 | 0.005 | [0.22–1.76] | 0.19 ± 3.28 | 1.000 | [−0.70 to 1.09] |
| | Kerr | Poortmans | 1.59 ± 0.20 | 0.000 | [1.05–2.13] | 11.84 ± 3.47 | 0.000 | [10.89–12.78] | 16.64 ± 4.26 | 0.000 | [15.47–17.80] |
| | Kerr | Matigka | −3.92 ± 0.74 | 0.000 | [−5.90 to −1.93] | 2.03 ± 6.74 | 0.022 | [0.19–3.86] | −11.07 ± 6.43 | 0.000 | [−12.83 to −9.32] |
| | Lee | Poortmans | 0.96 ± 0.19 | 0.000 | [0.43–1.49] | 10.85 ± 4.60 | 0.000 | [9.60–12.10] | 16.44 ± 5.70 | 0.000 | [14.88– 17.99] |
| | Lee | Matiegka | −4.54 ± 0.62 | 0.000 | [−6.22 to −2.87] | 1.04 ± 4.51 | 0.145 | [−0.19 to 2.27] | −11.27 ± 4.39 | 0.000 | [−12.47 to− 10.07] |
| | Poortmans | Matiegka | −5.51 ± 0.72 | 0.000 | [−7.45 to −3.57] | −9.81 ± 7.86 | 0.000 | [−11.95 to −7.68] | −27.71 ± 8.93 | 0.000 | [−30.15 to −25.27] |
| BM (kg) | Kerr | Rocha | −2.86 ± 0.88 | 0.000 | [−3.02 to −2.71] | −3.19 ± 1.04 | 0.000 | [−3.39 to 2.98] | −2.49 ± 0.97 | 0.000 | [−2.68 to −2.29] |
| BM (%) | Kerr | Rocha | −4.26 ± 1.27 | 0.000 | [−4.48 to −4.03] | −4.26 ± 1.51 | 0.000 | [−4.56 to −3.96] | −4.26 ± 1.66 | 0.000 | [−4.59 to −3.93] |
| MBI | Kerr/Kerr | Lee/Rocha | 1.02 ± 0.02 | 0.000 | [0.95–1.10] | 1.09 ± 0.34 | 0.000 | [1.01–1.19] | 0.94 ± 0.43 | 0.000 | [0.83–1.06] |
| | Kerr/Kerr | Poortmans/Rocha | 1.08 ± 0.02 | 0.000 | [1.01–1.16] | 1.11 ± 0.34 | 0.000 | [1.02–1.21] | 0.22 ± 0.58 | 0.002 | [0.06–0.38] |
| | Kerr/Kerr | Matiegka/Rocha | 0.73 ± 0.05 | 0.000 | [0.59–0.88] | 1.17 ± 0.48 | 0.000 | [1.04–1.30] | 1.06 ± 0.42 | 0.000 | [0.94–1.17] |
| | Lee/Rocha | Poortmans/Rocha | 0.06 ± 0.01 | 0.000 | [0.02–0.09] | 0.02 ± 0.14 | 1.000 | [−0.02 to 0.06] | −0.73 ± 0.27 | 0.000 | [−0.79 to −6.65] |
| | Lee/Rocha | Matiegka/Rocha | −0.28 ± 0.04 | 0.000 | [−0.39 to −0.18] | 0.07 ± 0.29 | 0.087 | [−0.01 to 0.15] | 0.11 ± 0.21 | 0.000 | [0.05–0.17] |
| | Poortmans/Rocha | Matiegka/Rocha | −0.34 ± 0.04 | 0.000 | [−0.47 to −0.22] | 0.06 ± 0.33 | 0.611 | [−0.04 to 0.146] | 0.84 ± 0.38 | 0.000 | [0.73–0.94] |

**Note:**
SD, standard deviation; 95%CI, 95% Confidence Interval; Max, maximum value; Min, minimum value; MM, Muscle Mass; BM, Bone Mass; MBI, Muscle/Bone index; kg, Kilograms; %, Percentages.

**Table 5 Lin's concordance correlation coefficient.**

| Variable | Formula | | General sample (*n* = 130) | | | Male sample (*n* = 71) | | | Female sample (*n* = 59) | | |
|---|---|---|---|---|---|---|---|---|---|---|---|
| | | | CCC | r | Cb | CCC | r | Cb | CCC | r | Cb |
| MM (kg) | Kerr | Lee | 0.96 | 0.97 | 0.98 | 0.90 | 0.97 | 0.94 | 0.86 | 0.91 | 0.95 |
| | Kerr | Poortmans | 0.96 | 0.97 | 0.98 | 0.91 | 0.96 | 0.95 | 0.84 | 0.89 | 0.93 |
| | Kerr | Matiegka | 0.47 | 0.91 | 0.51 | 0.33 | 0.75 | 0.44 | 0.13 | 0.82 | 0.16 |
| | Lee | Poortmans | 0.97 | 0.97 | 0.99 | 0.94 | 0.95 | 0.99 | 0.80 | 0.88 | 0.91 |
| | Lee | Matiegka | 0.50 | 0.92 | 0.54 | 0.48 | 0.80 | 0.60 | 0.09 | 0.81 | 0.12 |
| | Poortmans | Matiegka | 0.44 | 0.91 | 0.49 | 0.44 | 0.74 | 0.59 | 0.08 | 0.75 | 0.11 |
| MM (%) | Kerr | Lee | 0.86 | 0.87 | 0.98 | 0.79 | 0.85 | 0.93 | 0.67 | 0.68 | 0.99 |
| | Kerr | Poortmans | 0.86 | 0.90 | 0.95 | 0.82 | 0.87 | 0.94 | 0.63 | 0.74 | 0.84 |
| | Kerr | Matigka | −0.29 | −0.37 | 0.76 | −0.03 | −0.04 | 0.85 | 0.04 | 0.20 | 0.18 |
| | Lee | Poortmans | 0.87 | 0.90 | 0.96 | 0.87 | 0.88 | 0.99 | 0.62 | 0.71 | 0.87 |
| | Lee | Matiegka | −0.09 | −0.13 | 0.67 | 0.34 | 0.36 | 0.95 | 0.12 | 0.66 | 0.18 |
| | Poortmas | Matiegka | −0.18 | −0.29 | 0.63 | 0.27 | 0.28 | 0.97 | 0.05 | 0.30 | 0.15 |
| BM (kg) | Kerr | Rocha | 0.33 | 0.86 | 0.38 | 0.15 | 0.72 | 0.21 | 0.12 | 0.64 | 0.19 |
| BM (%) | Kerr | Rocha | 0.06 | 0.48 | 0.13 | 0.08 | 0.54 | 0.15 | 0.04 | 0.38 | 0.11 |
| MBI | Kerr/Kerr | Lee/Rocha | 0.13 | 0.71 | 0.19 | 0.10 | 0.68 | 0.15 | 0.05 | 0.41 | 0.13 |
| | Kerr/Kerr | Poortmans/Rocha | 0.15 | 0.72 | 0.21 | 0.11 | 0.68 | 0.16 | 0.07 | 0.48 | 0.14 |
| | Kerr/Kerr | Matiegka/Rocha | −0.12 | −0.39 | 0.32 | 0.02 | 0.18 | 0.09 | −0.11 | −0.16 | 0.70 |
| | Lee/Rocha | Poortmans/Rocha | 0.88 | 0.92 | 0.96 | 0.92 | 0.94 | 0.99 | 0.68 | 0.78 | 0.87 |
| | Lee/Rocha | Matiegka/Rocha | −0.10 | −0.16 | 0.68 | 0.44 | 0.49 | 0.89 | 0.07 | 0.52 | 0.14 |
| | Poortmans/Rocha | Matiegka/Rocha | −0.15 | −0.24 | 0.63 | 0.41 | 0.48 | 0.85 | 0.04 | 0.27 | 0.14 |

**Note:**
CCC, Concordance Correlation Coefficient; r, Pearson correlation coefficient; Cb, Bias correction factor; MM, Muscle Mass; BM, Bone Mass; MBI, Muscle/Bone index; kg, Kilograms; %, Percentages.

# DISCUSSION

The objectives of this study were: 1) to investigate the agreement between different MM and BM formulas, and the MBI, and to establish the differences between them, in a healthy young adult population; and 2) to analyze if there are differences between males and females in the comparison of MM, BM and MBI formulas. The research hypotheses were as follows: a) the different formulas for estimate MM, BM and MBI with anthropometry are not interchangeable; and b) sex exerts influence on the comparative analysis of these formulas. Considering the findings of the current study, the hypotheses can be accepted because formulas to estimate MM, BM and MBI are not interchangeable, as evidenced by the statistically significant differences observed among MM and BM estimation formulas, along with the MBI. Furthermore, finding support the hypothesis that sex significantly influences the comparative assessment of these formulas, as indicated by the significant sex effect observed in the divergences between formulas in the majority of parameters examined.

The present study showed significant differences between the values reported by all the formulas used to assess MM, and the lack of agreement between them in the general, male and female samples. Furthermore, concordance among equations was notably higher in
men, contrasting with lower levels in women. Only one previous study has been carried out on this issue in a sample of weightlifters, whose results largely agreed with those from the present findings (*Fernández-Vieitez & Aguilera, 2001*). However, this previous study had some limitations, such as having a very small sample size or not specifying the sex of the participants (*Fernández-Vieitez & Aguilera, 2001*), which, in the light of the results from the present research, could influence the comparability of the formulas. The differences found between the results reported by the formulas in the present investigation could be explained by the differences in the validation context. Along this line, Kerr's formula (*Kerr & Ross, 1991*) included a sample of 1,669 subjects of both sexes, aged between 6 and 77 years old with different levels of physical activity. In this formula, cadaver dissection was used as a means of comparison for validation of the formula. An advantage of this formula is that the anthropometric values were corrected according to the height of subject (*Kerr & Ross, 1991*), which could be essential when comparing subjects with different heights or a population undergoing growth (*Esparza-Ros & Vaquero-Cristóbal, 2023*). Regarding *Lee*'s *et al. (2000)* formula, it was validated in 324 sedentary individuals, of whom 244 were not obese, while 80 were obese. The formula was validated with magnetic resonance imaging (MRI). Paradoxically, despite the fact that the validation of this formula did not include the active population, it has been one of the most widely used in the assessment of the MM of athletes (*Alvero-Cruz et al., 2009*). With respect to Poortmans' formula (*Poortmans et al., 2005*), it was validated in only 59 individuals, 30 males and 29 females, aged between 7 and 24 years old. The formula was validated using Dual-energy X-ray absorptiometry (DXA) as validation method. Finally, with respect to *Matiegka (1921)* he validated his formula with only 12 young males. According to the present findings, the Matiegka equation demonstrated significantly higher values as compared to other formulas, particularly in the female group, which could have hindered the concordance results in the female group. Notably, the equation's author excluded women from the sample, raising concerns about its suitability for use in female populations. This finding raises important considerations regarding the equation's generalization, especially in contexts involving female populations. Therefore, in general terms, the different validation contexts in which these formulas were developed mean that the results reported by the different MM estimation formulas are not comparable.

Another finding of the present study was that differences were found between the formulas used to estimate BM, as well as a lack of concordance between the different formulas in the general sample and divided by sex. No previous studies have addressed this issue, despite the importance of BM assessment, given its influence on performance and health (*Kraemer et al., 2020*; *Clynes et al., 2021*). The differences in the results reported by the formulas could be due to differences in the validation context. Along this line, *Rocha (1975)* does not indicate the method used as the gold standard for validation. Thus, the differences also shown between the BM formulas might make the reported results not interchangeable.

In addition, the results from the present research indicated differences between all the MBI values, with no significant agreement between any of them for the general sample and separated by sex. The MBI parameter has been a novelty in recent years, and it has been

pointed out that it could be of great interest in the field of sports performance (*Esparza-Ros & Vaquero-Cristóbal, 2023*). However, few studies to date have included this parameter (*Bernal-Orozco et al., 2020*), and the results of the present study suggest that its use and interpretation need to take into consideration formulas used for the estimation of MM and BM prior to the calculation of this index.

Consequently, the present study represents the first investigation to assess, in a sample of significant size and with a known distribution of men and women, the comparability of MM estimation equations through anthropometric methods. Additionally, the current research Rhas been carried out on the influence of sex on this comparability, an aspect not explored until this research in the scientific literature. Lastly, the novelty of this study lies in its comprehensive analysis, not only of MM and MO but also of the index relating both masses, MBI, providing the first evidence of the comparability of both masses and their interrelationship, while not overlooking the influence of sex. This approach could mark the beginning of new lines of research dedicated to exploring this facet.

Concerning the practical implementation of this study, the use of anthropometry in practical contexts such as gyms and fitness centers is very common, making the comparison of results obtained by different estimation equations highly relevant, especially given the scarcity of research on this matter. The present study underscores the imperative of consistently employing the same anthropometric equation when evaluating MM, BM, and MBI during an individual's progression or in comparative analyses with a reference population. It is paramount to recognize that comparing outcomes obtained from different equations could yield notable inaccuracies that could affect the interpretation of MM, BM, and MBI data by professionals using anthropometry in clinical or research settings.

Regarding the limitations of this study, the present investigation did not analyze a method that could be used as a gold standard for the assessment of MM and BM. Thus, it is not possible to know which formula is the most valid, although this could be a future line of research. Another limitation of the present investigation is that because the development of muscle and bone mass could differ between subjects, the data cannot be extrapolated to other populations such as obese individuals, individuals with eating disorders, or individuals in other age ranges. Therefore, similar studies should be performed to assess outcomes in these specific population groups, using a gold standard to evaluate the validity of the findings, and to evaluate which equation should be used, once it has been found that the equations are not interchangeable.

In light of the results of the present study, future research avenues should not only analyze the differences found in anthropometry for the estimation of MM and BM but also explore the comparability of these results with different estimation methods such as DXA, MRI, bioelectrical impedance, 3D scanner analysis, among others; determining sex differences in this respect. A second line of development would be, once it is known that the anthropometry formulas are not comparable, to analyse which of them could be more valid with respect to the gold standard for muscle mass, *i.e.* MRI (*Costa-Moreira et al., 2015*), and for bone mass, *i.e.* with respect to DXA (*Costa-Moreira et al., 2015*), and the influence of sex on these issues.

## CONCLUSIONS

In the present investigation, significant differences were found between all the values reported by the MM, BM, and MBI formulas in the general sample and divided by sex, without a significant agreement found between these formulas in recreational strength trainees. These findings highlight that when assessing the MM, BM or MBI of recreational strength trainees based on anthropometric variables, to compare their results with reference tables, other studies, or data from other athletes, or to analyze their evolution over time, extreme caution must be taken to always use the same prediction equation as those tables, studies, reference data or previous data. Otherwise, the comparisons would not be equivalent. Therefore, whenever an anthropometric assessment of both MM and BM is performed, the formula used must be clarified. Furthermore, in the absence of future research analyzing the validity of these equations against a reference method, in the light of the present research it could be recommended to use the Kerr equation both for the estimation of MM, BM and MBI in recreational strength trainers. This recommendation is based on the fact that almost all formulas differ in their result and at least Kerr's strategy follows the same line for the analysis of muscle and bone mass, it was validated by using the cadaver dissection as a comparison method, it adds a correction for height when estimating body composition and it included for validation a sufficient sample of male and female subjects and active subjects.

## ACKNOWLEDGEMENTS

The authors would like to thank all the participants and helpers who made this research possible. This study is part of the doctoral thesis of Nicolás Baglietto.

### Funding
The authors received no funding for this work.

### Competing Interests
The authors declare that they have no competing interests.

### Author Contributions
- Nicolás Baglietto performed the experiments, prepared figures and/or tables, authored or reviewed drafts of the article, and approved the final draft.
- Mario Albaladejo-Saura analyzed the data, prepared figures and/or tables, authored or reviewed drafts of the article, and approved the final draft.
- Francisco Esparza-Ros conceived and designed the experiments, authored or reviewed drafts of the article, and approved the final draft.
- Raquel Vaquero-Cristóbal conceived and designed the experiments, analyzed the data, authored or reviewed drafts of the article, and approved the final draft.

## Human Ethics

The following information was supplied relating to ethical approvals (*i.e.*, approving body and any reference numbers):

The Catholic University of Murcia granted Ethical approval to carry out the study within its facilities (Ethical Application: CE072103).

## Data Availability

The raw measurements are available in the Supplemental File 1.

## Supplemental Information

Supplemental information for this article can be found online at http://dx.doi.org/10.7717/peerj.17506#supplemental-information.

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
