# Peer review of "Agreement and differences between the equations for estimating muscle and bone mass using the anthropometric method in recreational strength trainees"

_PeerJ, doi:10.7717/peerj.17506_

## Round 0.1 · original submission · Minor Revisions

The authors should clarify issues pointed out by the reviewers. Please, be careful about methods to guarantee that all study procedures are clear and provided. Furthermore, discussion should be improved, and practical applications should be highlighted.

·

Basic reporting

The aim of the present study was to compare the values of MM and BM obtained through different anthropometric formulas in healthy and active adult recreational strength trainees. The manuscript is well structured methodologically, but the research gap is not very robust and some sections need further revision.

Experimental design

Please clarify the study design: is it a retrospective or prospective observational study? Also, indicate the reference of the STROBE guidelines (in this case the specific scale for observational studies). The participants section should be placed before the study design (lines 92-100). This section should only contain the characteristics of the sample and the ethical procedures (lines 87-91). The study design section should only refer to the experimental approach used; the methodological procedures associated with ISAK should be referred to later (as procedures and then variables). After the Participants (first place) and study design (second place) sections, the authors should include information on the inclusion and exclusion criteria in the Selection criteria subsection (lines 100-119).

Validity of the findings

The results are valid and robust, but some points should be specified. In particular, p-values cannot be presented in ranges (e.g. p=0.001-0.000, lines 201); please specify for each variable or put as p<0.01, p<0.05 or p<0.001. Also, it should describe the other inferences. For example, for ANOVA it should be: F statistics, eta squared, post hocs).

Additional comments

I would like to leave you with some additional comments on the study, section by section::
- Abstract/Introduction: Please, standardize the research aims (lines 24-26, 74-78).
- Methods: Regarding the second objective (i.e., to analyze if there are differences between males and females in the comparison of MM, BM and MBI formulas), what formulas are actually proposed? The methodology only refers to the ISAK guidelines and the results are not very clear.
- Disucssion: the first paragraph of the discussion should clarify the confirmation and/or rejection of the hypotheses by research aim (lines 227-230). In addition to the research limitations, the authors should present future perspectives and practical applications (lines 282-290).
- The authors should refer specifically to the forms analyzed, and the comparison with MRI and DXA was not made in this study.Although an indirect comparison with the gold standards may be presented, the discussion should be written in a logical manner Although an indirect comparison with the gold standards may be presented, the discussion should be written in accordance with the logic of the presentation of the results.
- Conclusions: In the conclusion section, it should be clear which equation is most appropriate for estimating muscle and bone mass using anthropometrics in recreational strength exercisers.
- In the tables, abbreviations are missing from the legend (i.e., CCC, Cb, partial eta squared, etc.).

·

Basic reporting

This article aims to compare the values of MM and BM obtained through different anthropometric formulas in healthy and active adult recreational strength trainees.
The text adheres to the criteria of clarity and unambiguous, professional English usage throughout. It provides sufficient background and context by referencing relevant literature and field knowledge. Additionally, the article maintains a professional structure with clear sections, including figures, tables, and shared raw data, enhancing clarity and facilitating understanding for readers.

Experimental design

The article meets the criteria for original primary research within the aims and scope of the journal, since
it addresses a well-defined and meaningful research question regarding (compairing the values of MM and BM obtained through different anthropometric formulas),

The article investigates the outcome of different anthropometric equations for estimating components of body composition such as lean mass or bone mass. The use of anthropometry in practical contexts such as gyms and fitness centers is very common, making the comparison of results obtained by different estimation equations highly relevant, especially given the scarcity of research on this matter. Thus, the article presents an innovative and well-framed research problem, which is well-defined and aligned with the existing literature. The methodology used in the research design, participant recruitment, anthropometric evaluation, and statistical analysis is appropriate. Ethical procedures are described adequately, although it should be specifically mentioned that participants signed an informed consent. The results are presented clearly in easily readable tables.

Validity of the findings

The discussion of the results is appropriately thorough for research of this type, providing a clear interpretation for each type of result based on robust and adequate statistical treatment. The conclusions drawn by the authors are well-established and coherent with both the theoretical foundation and the presented research problem. Due to the lack of literature on this specific topic, the discussion does not extensively compare with existing articles. It would be beneficial for the authors to emphasize the practical applications of their research.

Additional comments

The article investigates a relevant topic using appropriate methodology and presents the results clearly, so I believe it should be published.

---

## Round 0.2 · Minor Revisions

The authors have addressed all the reviewers concerns and the manuscript has substantially improved. Considering that a Table was included (Table 1), I would suggest if it is possible to include the specific equations in the Table, instead of only providing the names. Please, consider to include this at this stage.

·

Basic reporting

After the first peer review, the article has undergone significant and structuring changes at all stages of the research. The research gap has been clarified, and the authors have reformulated the methodology and the presentation and interpretation of the results. The manuscript is now more robust.

Experimental design

The research aim and scope are well defined, revealing and meaningful. Also, the ethical and methodological procedures have been thoroughly reformulated in line with the recommendations issued.

Validity of the findings

The results are adequately described. In the revised version, the authors have clarified the presentation of p-values and effect sizes.

Additional comments

Nothing more to report.

·

Basic reporting

.

Experimental design

.

Validity of the findings

.

Additional comments

The authors have provided clear and comprehensive responses to all queries raised by the reviewers, demonstrating a thorough understanding of the feedback received. Based on the thoroughness of their responses, I am of the opinion that the article is now suitable for publication.

---

## Round 0.3 · accepted · Accept

The authors have addressed all the reviewers' comments and, in my opinion, it meets the standards for publication.